# Comparison of the MultiViewScope Stylet Scope and the direct laryngoscope with the Miller blade for the intubation in normal and difficult pediatric airways: A randomized, crossover, manikin study

Kohei Godai📷*, Takahiro Moriyama, Yuichi Kanmura

Department of Anesthesiology and Critical Care Medicine, Graduate School of Medical and Dental Sciences, Kagoshima University, Kagoshima, Japan

* kxg179@icloud.com

## Abstract

**Data Availability Statement:** All data are available from Figshare (doi: 10.6084/m9.figshare.

### Background

Managing difficult pediatric airway is challenging. The MultiViewScope (MVS) Stylet Scope is reported to be useful in difficult pediatric airway. In this randomized crossover study, we compared the effectiveness of the MVS Stylet Scope to a standard direct laryngoscope with Miller #1 blade in simulated normal and difficult airways.

### Methods

Fifteen expert anesthesiologists and Fifteen anesthesiology residents participated in the study. Participants were asked to perform intubation with the Airsim Baby manikin first, and then with the Airsim Pierre Robin manikin. Participants in each group used the intubation devices in a randomized order. The primary outcome was the time of successful intubation. The secondary outcomes were the force exerted on the incisors during intubation, Cormack–Lehane scale, the difficulty of intubation.

### Results

There were no differences between MVS Stylet Scope and Direct laryngoscope in the time of successful intubation by the expert anesthesiologists or the anesthesiology residents in a normal or difficult pediatric airway. MVS Stylet Scope significantly improved the force exerted on the incisors during intubation in the expert anesthesiologists or the anesthesiology residents in a normal or difficult pediatric airway. MVS Stylet Scope significantly improved Cormack–Lehane scale, and the difficulty of intubation with difficult pediatric airway situation in both expert anesthesiologists and anesthesiology residents.

11591298.v1 and doi: 10.6084/m9.figshare.
11604024.v1).

**Funding:** The authors received no specific funding
for this work.

**Competing interests:** The authors have declared
that no competing interests exist.

## Conclusions

Although less forces on the incisors and improved view of glottis were observed with the
MVS Stylet Scope, MVS Stylet Scope did not shorten the time of intubation. The results of
this study mean that the MVS Stylet Scope may be a less invasive airway devise than the
direct laryngoscope with the Miller blade in the pediatric airway management. For the next
step, we need to evaluate the MVS Stylet Scope in the real patients as an observational
study.

## Introduction

Managing difficult pediatric airway is challenging especially under two years of age, because of
the following physiological features [1]. Neonates and infants have elevated metabolic rate and
lower functional residual capacity compared with adults [2]. Neonates and infants become
desaturated more rapidly than adults because of the physiological features. Awake intubation
is recommended when managing anticipated difficult airway in adults [3–5]. Awake intuba-
tion is rarely possible in children due to lack of cooperation of pediatric patients [6]. Conduct-
ing randomized clinical trials in children with difficult airways is very difficult. The major
challenges to performing such trials are the limited number of subjects qualifying for enroll-
ment and variability in airway anatomy.

The MultiViewScope (MVS) is a video laryngoscope system, in which the video monitor
handle can be attached to a stylet scope, Miller blade, Macintosh blade, or fiberscope. MVS
Stylet Scope is reported to be useful in difficult pediatric airway associated with Schwartz–Jam-
pel syndrome [7]. MVS Stylet Scope has a rigid fiberoptic device with a 90-degree curved tip,
which facilitates targeted intubation. MVS Stylet Scope is different from StyletScope (Nihon
Kohden, Tokyo, Japan), which has the 75-degree flexible distal tip [8].

The objective of this randomized crossover study was to determine which of the devices, a
standard direct laryngoscope with Miller #1 blade, or the MVS Stylet Scope, was associated
with shorter times for successful tracheal intubation by expert anesthesiologists or anesthesiol-
ogy residents in two manikins, simulating infants with a normal airway or the difficult airway
of Pierre Robin Sequence.

## Materials and methods

This manuscript adheres to the applicable CONSORT guidelines. The supporting CONSORT
checklist is available as S1 Fig. The ethics committee of Kagoshima University Hospital
approved this randomized, crossover, manikin study. We conducted this study from Septem-
ber 2018 to December 2019 at Kagoshima University Hospital. We prospectively registered
this study on a publicly accessible database (UMIN Clinical Trials Registry ID:
UMIN000033456). We obtained written informed consent from all participants. Fifteen Japa-
nese Society of Anesthesiology board certified anesthesiologists as expert anesthesiologists and
Fifteen anesthesiology residents participated in the study. We decided to analyze data sepa-
rately from expert anesthesiologists and anesthesiology residents to evaluate utility of the MVS
Stylet Scope in both groups.

### Protocol

We asked all participants about their experience in anesthesiology, in pediatric anesthesia, and
in MVS Stylet Scope. All participants received a 10-min standardized demonstration before

the measurements. All participants were given 10 min to practice on a 3–6-month Airsim Baby manikin (TruCorp, Co.Armagh, N. Ireland) with MVS stylet scope. We used Airsim Baby manikin as normal pediatric airway. A 3–6-month Airsim Pierre Robin manikin (TruCorp, Co.Armagh, N. Ireland), which had been designed in line with the characteristics of an infant patient with Pierre Robin syndrome, was used as the difficult pediatric airway. The study devices consisted of the MVS Stylet Scope, MVS-SC25 (MPI, Tokyo, Japan, Fig 1) and the direct laryngoscope with the size 1 blade of Miller (HEINE Optotechnik, Herrsching, Germany).

The outer diameter of MVS-SC25 was 2.5mm. The endotracheal tube of inner diameter of 3.0 to 4.0 mm were able to be attached to the MVS-SC25. We used a 3.5 mm uncuffed endotracheal tube (Covidien, Medtronic, Minneapolis, MN, USA) for each intubation attempt. We used a 6 French outer diameter malleable intubating stylet (Parker Medical, Bridgewater, CT, USA) for intubation with the direct laryngoscope with the Miller blade. We standardized the stylet shape to mimic the curve of the MVS Stylet Scope. Participants performed intubation with the Airsim Baby manikin first, and then with the Airsim Pierre Robin manikin. We did not randomize the order of manikins, because we compared the devices not the manikins. Participants performed intubation twice (with the MVS Stylet Scope or with the direct laryngoscope) with each manikin. Participants in each group used the intubation devices in an order randomized by the internet-based software [Research Randomizer (Version 4.0) Retrieved on October 13, 2016, from http://www.randomizer.org/]. There was no restriction of randomization (such as blocking and block size). The researcher (KG) generated the random allocation sequence, enrolled participants, and assigned participants to interventions. No one is blinded to the allocation. The duration of successful intubation (between the time when endotracheal tube entered the oral cavity and the time when the lungs were positively ventilated) was measured by the same researcher (KG). We considered it is an unsuccessful attempt, if intubation could not be completed in 90 s. We considered it is an unsuccessful intubation, if intubation could not be performed on the third attempt. Participants assessed the best glottic view on the Cormack–Lehane scale, the difficulty of intubation (NRS, Numerical Rating Scale of 0–10, where 0 indicates "no difficulty" and 10 indicates "maximum difficulty"), and preference of the two devices (the MVS Stylet Scope or the direct laryngoscope) for intubation. A pressure film transducer (LLLW Prescale Pressure Film, full scale 0.6 MPa, accuracy 10%, Fujifilm, Tokyo, Japan) was used to measure the force exerted on the incisors during intubation. The film transducer was composed of two layers. One contained microcapsules full of a coloring fluid substance; the other one was the fixing layer. When the microcapsules broke the films underwent a color change proportional to the applied pressure. After each intubation, the impressed layer was scanned and processed using Image J (U. S. National Institutes of Health, Bethesda, Maryland, USA), which generated a matrix containing the mean pressure intensity with a spatial resolution of $0.1 \text{ mm}^2$. The intensity of the resultant force was then calculated by adding the contributions in pressure on each $\text{mm}^2$ of the film transducer.

## Statistical analysis

The primary outcome was the time of successful intubation. The secondary outcomes were the force exerted on the incisors during intubation, Cormack–Lehane scale, the difficulty of intubation, preference of the two devices for intubation, and failure rate [9]. We considered 10 sec in the time of successful intubation as clinically significant. To detect a difference of 10 sec in the time of successful intubation with a two-sided approximation accepting an α error of 5% and a β error of 10%, the required study size was calculated as 15 participants in one group based on a previous study using Power and Sample Size Calculation version 3.1.2 (Dupont

# MVS Stylet Scope, OD 2.5mm

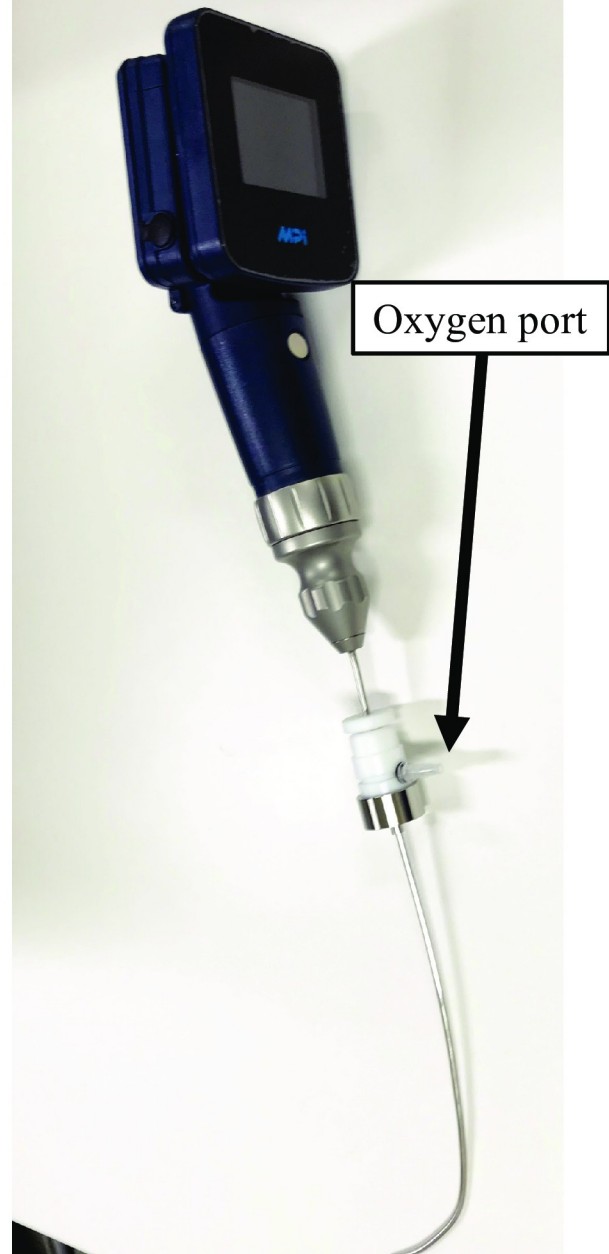

Oxygen port

Abbreviations: MVS, MultiViewScope; OD, Outer Diameter

**Fig 1. The MVS Stylet Scope.**

WD and Plummer WD, Vanderbilt University, Nashville, TN) [10]. Normally distributed data are shown in means and standard deviations (SD). Data, which were not normally distributed, are shown in medians and interpercentile ranges (IQR). Statistical analysis was performed using Mann-Whitney test and 2-way ANOVA and ANOVA for Cross-over design (GraphPad Prism 7.0, La Jolla, CA, USA). $P < 0.05$ was considered statistically significant except for

carryover effects. For carryover effects, we considered $P < 0.1$ as statistically significant. Comparison of data from study period 1 were performed, when carryover effect was detected. All data and raw images were deposited in a public repository [11, 12].

## Results

The CONSORT diagram is shown in Fig 2.

Of 30 participants considered eligible for the study, no participant was excluded. The data from all participants were included in the study. Table 1 shows the participants' characteristics. There was no clinically significant difference between groups. There were no missing data.

The primary and secondary outcomes are shown in Tables 2 to 5.

The detailed analyzed data are shown in S1–S4 Tables. MVS Stylet Scope significantly improved the force exerted on the incisors during intubation in the expert anesthesiologists with normal pediatric airway (Table 2). There were no differences between MVS Stylet Scope and Direct laryngoscope in the time of successful intubation, Cormack–Lehane scale, and the difficulty of intubation in the expert anesthesiologists with normal pediatric airway. Eight out of 15 preferred MVS Stylet Scope and there was no intubation failure.

MVS Stylet Scope significantly improved the force exerted on the incisors during intubation, Cormack–Lehane scale, and the difficulty of intubation in the expert anesthesiologists with difficult pediatric airway (Table 3). There were no differences between MVS Stylet Scope and Direct laryngoscope in the time of successful intubation in the expert anesthesiologists with difficult pediatric airway. Fourteen out of 15 preferred MVS Stylet Scope and there was no intubation failure.

We analyzed data from study period 1, because of carryover effect in the force in the anesthesiology residents with normal pediatric airway (Table 4). MVS Stylet Scope significantly improved the force exerted on the incisors during intubation in the anesthesiology residents with normal pediatric airway. There were no differences between MVS Stylet Scope and Direct laryngoscope in the time of successful intubation, Cormack–Lehane scale, and the difficulty of intubation in the anesthesiology residents with normal pediatric airway. Eight out of 15 preferred MVS Stylet Scope and there was no intubation failure.

MVS Stylet Scope significantly improved the force exerted on the incisors during intubation, Cormack–Lehane scale, and the difficulty of intubation in the anesthesiology residents with difficult pediatric airway (Table 5). There were no differences between MVS Stylet Scope and Direct laryngoscope in the time of successful intubation in the anesthesiology residents with difficult pediatric airway. Twelve out of 15 preferred MVS Stylet Scope and there was no intubation failure.

The subject profile plots are shown in S2–S5 Figs.

## Discussion

MVS Stylet Scope did not shorten time of successful intubation in any situation compared with direct laryngoscope using Miller #1 blade. MVS Stylet Scope, however, did improve the force exerted on the incisors during intubation of normal pediatric airway in both expert anesthesiologists and anesthesiology residents. In addition, MVS Stylet Scope significantly improved the force exerted on the incisors during intubation, Cormack–Lehane scale, and the difficulty of intubation with difficult pediatric airway situation in both expert anesthesiologists and anesthesiology residents. Although the differences of Cormack–Lehane scale between the MVS Stylet Scope and the direct laryngoscope using Miller #1 blade were small, we consider the differences are clinically significant. The reason is that the participants in the both groups reported improved difficulty of intubation with the MVS Stylet Scope.

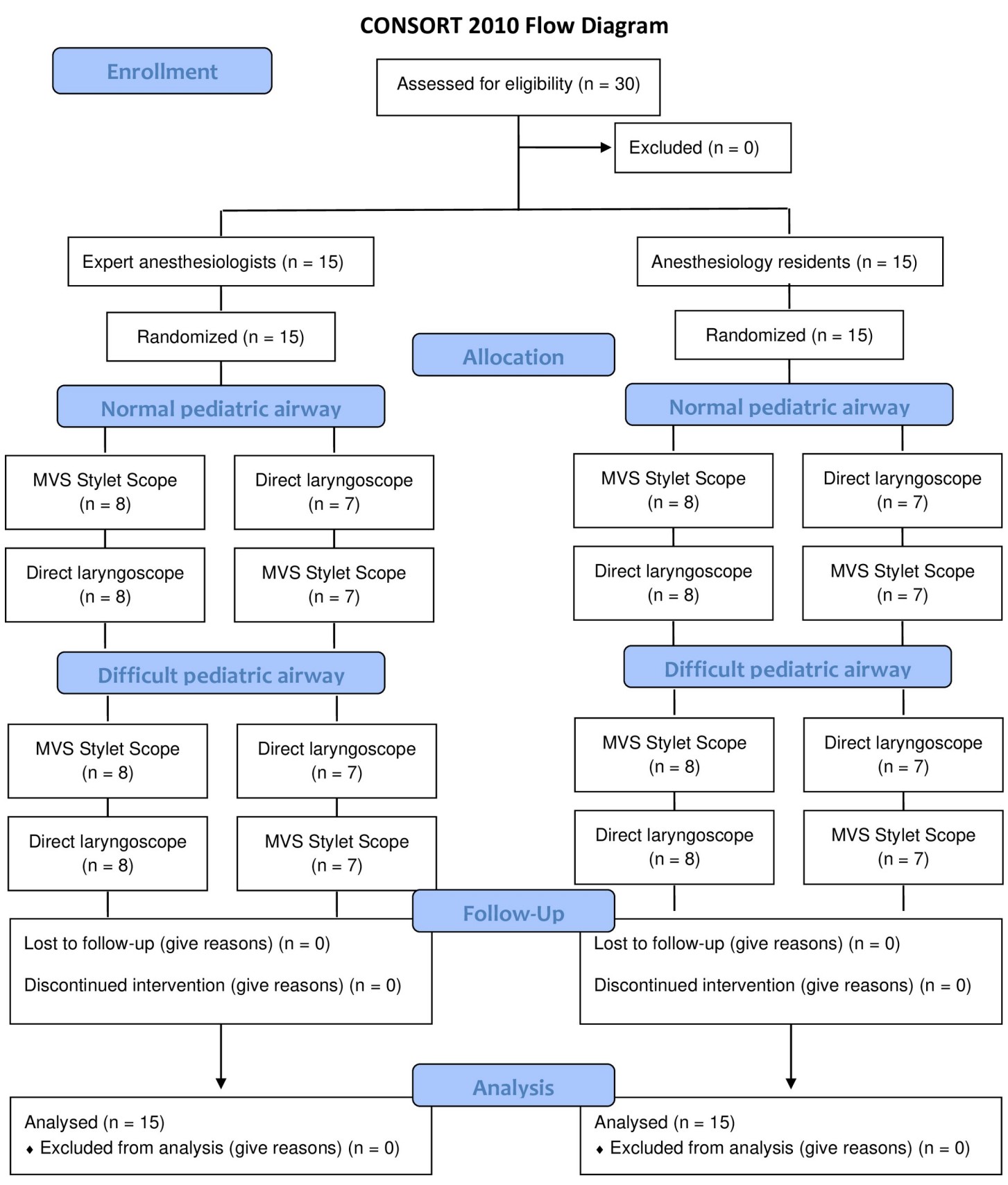

**Fig 2. The CONSORT flow diagram of the study.**

**Table 1. Participants' characteristics.**

| Expert anesthesiologists | SS then DL (n = 8) | DL then SS (n = 7) |
|---|---|---|
| Experience in anesthesia | | |
| Less than a year (n) | 0 | 0 |
| 1–3 years (n) | 0 | 0 |
| 3–5 years (n) | 1 | 0 |
| More than 5 years (n) | 7 | 7 |
| Experience in PA | | |
| Less than a year (n) | 0 | 0 |
| 1–3 years (n) | 0 | 0 |
| 3–5 years (n) | 1 | 1 |
| More than 5 years (n) | 7 | 6 |
| Experience in using SS | | |
| For the first time (n) | 6 | 7 |
| Less than 10 times (n) | 2 | 0 |
| More than 10 times (n) | 0 | 0 |
| Anesthesiology residents | SS then DL (n = 8) | DL then SS (n = 7) |
| Experience in anesthesia | | |
| Less than a year (n) | 4 | 3 |
| 1–3 years (n) | 3 | 2 |
| 3–5 years (n) | 1 | 2 |
| More than 5 years (n) | 0 | 0 |
| Experience in PA | | |
| Less than a year (n) | 5 | 3 |
| 1–3 years (n) | 2 | 3 |
| 3–5 years (n) | 1 | 1 |
| More than 5 years (n) | 0 | 0 |
| Experience in using SS | | |
| For the first time (n) | 7 | 7 |
| Less than 10 times (n) | 1 | 0 |
| More than 10 times (n) | 0 | 0 |

Abbreviations: DL, Direct laryngoscope; PA, pediatric anesthesia; SS, MultiViewScope Stylet Scope.

**Table 2. Results of expert anesthesiologists with normal pediatric airway.**

| | Intubation device | | |
|---|---|---|---|
| | **SS (n = 15)** | **DL (n = 15)** | **SS—DL (n = 15)** |
| Time (sec), mean (SD) | 22.6 (8.8) | 24.3 (7.7) | |
| LSmean (95%CI), p value | | | 1.5 (-3.2 to 6.3), P = 0.50 |
| Force (N), mean (SD) | 33.4 (17.5) | 68.7 (19.5) | |
| LSmean (95%CI), p value | | | 34.8 (20.4 to 49.2), P < 0.001 |
| Cormack–Lehane scale (grade), median (IQR) | 1 (1 to 1) | 1 (1 to 1) | |
| LSmean (95%CI), p value | | | 0.13 (-0.07 to 0.33), P = 0.18 |
| Difficulty of intubation (NRS), median (IQR) | 2 (0 to 3) | 2 (0 to 4) | |
| LSmean (95%CI), p value | | | 0.38 (-0.78 to 1.53), P = 0.50 |

Abbreviations: CI, Confidence interval; DL, Direct laryngoscope; IQR, Interquartile range; LSmean, Least square mean; NRS, Numerical Rating Scale; SD, Standard deviation; SS, MultiViewScope Stylet Scope. *P* values were calculated using ANOVA for Cross-over design.

**Table 3. Results of expert anesthesiologists with difficult pediatric airway.**

|  | Intubation device | | |
|---|---|---|---|
|  | SS (n = 15) | DL (n = 15) | SS—DL (n = 15) |
| Time (sec), mean (SD) | 25.3 (13.2) | 28.0 (15.0) |  |
| LSmean (95%CI), *p* value |  |  | 2.9 (-6.1 to 12.0), P = 0.49 |
| Force (N), mean (SD) | 35.3 (28.2) | 55.8 (23.6) |  |
| LSmean (95%CI), *p* value |  |  | 20.7 (6.9 to 34.4), P = 0.006 |
| Cormack–Lehane scale (grade), median (IQR) | 1 (1 to 1) | 2 (1 to 2) |  |
| LSmean (95%CI), *p* value |  |  | 0.58 (0.35 to 0.82), P < 0.001 |
| Difficulty of intubation (NRS), median (IQR) | 2 (1 to 3) | 3 (3 to 5) |  |
| LSmean (95%CI), *p* value |  |  | 1.59 (0.57 to 2.60), P = 0.005 |

Abbreviations: CI, Confidence interval; DL, Direct laryngoscope; IQR, Interquartile range; LSmean, Least square mean; NRS, Numerical Rating Scale; SD, Standard deviation; SS, MultiViewScope Stylet Scope. *P* values were calculated using ANOVA for Cross-over design.

Although MVS Stylet Scope significantly improved the visualization of vocal cords and the difficulty of intubation in difficult pediatric airway manikin. The results were in line with the previous study [13]. Vlatten et.al. compared Bonfils retromolar intubation fiberscope with direct laryngoscope using Miller blade in the simulated difficult pediatric airway. The Bonfils retromolar intubation fiberscope is another type of optical stylets [14]. The Bonfils retromolar intubation fiberscope provided a better view of the larynx than direct laryngoscopy, but the time to intubate was not improved. Vlatten reported that median time to intubate with the Bonfils retromolar intubation fiberscope was 11 s. Mean time to intubate with MVS Stylet Scope was 20.2 to 31.6 s in our study. Other trials have reported that mean or median intubation time with Bonfils retromolar intubation fiberscope was 36 to 58 s [15–17]. The difference between MVS Stylet Scope and Bonfils retromolar intubation fiberscope is the tip designs. The MVS Stylet Scope has a 90-degree semi-malleable tip, although Bonfils retromolar intubation fiberscope has a 40-degree rigid tip.

**Table 4. Results of anesthesiology residents with normal pediatric airway.**

|  | Intubation device | | |
|---|---|---|---|
|  | SS (n = 15) | DL (n = 15) | SS—DL (n = 15) |
| Time (sec), mean (SD) | 28.1 (8.2) | 30.7 (10.8) |  |
| LSmean (95%CI), *p* value |  |  | 2.8 (-2.1 to 7.8), P = 0.24[a] |
| Cormack–Lehane scale (grade), median (IQR) | 1 (1 to 1) | 1 (1 to 1) |  |
| LSmean (95%CI), *p* value |  |  | 0.20 (-0.03 to 0.44), P = 0.08[a] |
| Difficulty of intubation (NRS), median (IQR) | 2 (1 to 3) | 3 (1 to 5) |  |
| LSmean (95%CI), *p* value |  |  | 1.13 (-0.30 to 2.57), P = 0.11[a] |
| Carryover effect |  |  |  |
| Force (N), mean (95%CI), Paired analysis | - | - | 62.3 (29.9 to 94.6), P = 0.005[a] |
| Comparison of data from study period 1 | MVS Stylet Scope | Direct laryngoscope | Mann-Whitney test |
| Force (N), median (IQR) | 17.6 (12.1 to 21.1) | 83.8 (28.1 to 106.9) | P = 0.006[b] |
| Force (N), n | 8 | 7 |  |

Abbreviations: CI, Confidence interval; DL, Direct laryngoscope; IQR, Interquartile range; LSmean, Least square mean; NRS, Numerical Rating Scale; SD, Standard deviation; SS, MultiViewScope Stylet Scope.

[a]*P* values were calculated using ANOVA for Cross-over design

[b]*P* value was calculated using Mann-Whitney test.

**Table 5. Results of anesthesiology residents with difficult pediatric airway.**

| | Intubation device | | |
|---|---|---|---|
| | **SS (n = 15)** | **DL (n = 15)** | **SS—DL (n = 15)** |
| Time (sec), mean (SD) | 29.7 (10.0) | 28.4 (9.1) | |
| LSmean (95%CI), *p* value | | | -1.2 (-5.8 to 3.5), *P* = 0.60 |
| Force (N), mean (SD) | 28.4 (20.1) | 50.1 (32.2) | |
| LSmean (95%CI), *p* value | | | 21.8 (8.7 to 34.8), *P* = 0.003 |
| Cormack–Lehane scale (grade), median (IQR) | 1 (1 to 1) | 1 (1 to 2) | |
| LSmean (95%CI), *p* value | | | 0.41 (0.13 to 0.69), *P* = 0.007 |
| Difficulty of intubation (NRS), median (IQR) | 2 (1 to 3) | 4 (2 to 6) | |
| LSmean (95%CI), *p* value | | | 1.47 (0.47 to 2.47), *P* = 0.007 |

Abbreviations: CI, Confidence interval; DL, Direct laryngoscope; IQR, Interquartile range; LSmean, Least square mean; NRS, Numerical Rating Scale; SD, Standard deviation; SS, MultiViewScope Stylet Scope. *P* values were calculated using ANOVA for Cross-over design.

We used pressure film transducer to measure the forces on the maxilla [18]. Some studies have used subjective measurement of the forces by a single observer [19, 20]. We believe that the film transducer method measures the forces more objectively than the subjective measurement. The mean measured forces were 17.6 to 72.8 N. The forces are comparable with previous reports [18, 21].

It is noteworthy that the times of successful intubation and Cormack–Lehane scale are similar in the two manikins. The mean time of successful intubation in the difficult pediatric airway was 28.4 sec, which was shorter than that of the normal pediatric airway (30.7 sec) in the anesthesiology residents. This may be due to the learning effects, because the mean time of successful intubation in the difficult pediatric airway (28.0 sec) was longer than that of the normal pediatric airway (24.3 sec) in the experienced expert anesthesiologists. Hippard et al. have reported that the time to intubation in the difficult pediatric airway manikin was shorter than the normal pediatric airway manikins [20]. We do not consider the learning effects have influenced our results, because we compared the devises, not the manikins.

The strength of this study are the rigorous randomized crossover design and objective measurement of the forces. Some airway manikin studies using have not applied randomized sequences, or evaluated the carryover effects [13, 18]. As we detected carryover effects in the anesthesiology residents with normal pediatric airway situation, the sequences may have effects on the outcomes.

Our study has several limitations. First, we used pediatric airway manikins. Some anesthesiologists believe that rigid plastic manikins, lack of collapsible soft tissues, absence of secretions make them unlikely to be useful surrogates for difficult airway [14]. The both manikins, which we used, have been evaluated in clinical studies repeatedly [20, 22, 23]. Another limitation of this study was that we did not randomize the order of manikins. This may account for our result that the time of successful intubation in the difficult pediatric airway was short compared to the normal pediatric airway in the anesthesiolosy residents. Third, most participants did not have experience using MVS Stylet Scope. Only three participants have used MVS Stylet Scope less than 10 times, and no one has used it more than 10 times. It is possible that the results might be different if experienced anesthesiologists participated.

In conclusion, although less forces on the incisors and improved view of glottis were observed with the MVS Stylet Scope, MVS Stylet Scope did not shorten the time of intubation. The results of this study mean that the MVS Stylet Scope may be a less invasive airway devise

than the direct laryngoscope with the Miller blade in the pediatric airway management. For the next step, we need to evaluate the MVS Stylet Scope in the real patients as an observational study.

## Supporting information

**S1 Fig. CONSORT checklist.**
(PDF)

**S2 Fig. Subject profile plots, expert anesthesiologists with normal pediatric airway.**
(PDF)

**S3 Fig. Subject profile plots, expert anesthesiologists with difficult pediatric airway.**
(PDF)

**S4 Fig. Subject profile plots, anesthesiology residents with normal pediatric airway.**
(PDF)

**S5 Fig. Subject profile plots, anesthesiology residents with difficult pediatric airway.**
(PDF)

**S1 Table. Detailed data of the results of expert anesthesiologists with normal pediatric airway.**
(PDF)

**S2 Table. Detailed data of the results of expert anesthesiologists with difficult pediatric airway.**
(PDF)

**S3 Table. Detailed data of the results of anesthesiology residents with normal pediatric airway.**
(PDF)

**S4 Table. Detailed data of the results of anesthesiology residents with difficult pediatric airway.**
(PDF)

## Acknowledgments

We would like to thank Editage (www.editage.com) for English language editing.

## Author Contributions

**Conceptualization:** Kohei Godai, Yuichi Kanmura.

**Data curation:** Kohei Godai.

**Formal analysis:** Kohei Godai.

**Funding acquisition:** Yuichi Kanmura.

**Investigation:** Kohei Godai.

**Methodology:** Kohei Godai.

**Project administration:** Kohei Godai.

**Resources:** Kohei Godai.

**Software:** Kohei Godai.

**Supervision:** Takahiro Moriyama, Yuichi Kanmura.

**Validation:** Takahiro Moriyama, Yuichi Kanmura.

**Visualization:** Kohei Godai.

**Writing – original draft:** Kohei Godai.

**Writing – review & editing:** Takahiro Moriyama, Yuichi Kanmura.

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
