## [Decision Letter · Decision Letter 0]

15 Jul 2020

PONE-D-20-17747

Comparison of the MultiViewScope Stylet Scope and the direct laryngoscope with the Miller blade for the intubation in normal and difficult pediatric airways: a randomized, crossover, manikin study

PLOS ONE

Dear Dr. Kohei Godai,

Thank you for submitting your manuscript to PLOS ONE. After careful consideration, we feel that it has merit but does not fully meet PLOS ONE’s publication criteria as it currently stands. Therefore, we invite you to submit a revised version of the manuscript that addresses the points raised during the review process.

We look forward to receiving your revised manuscript.

Kind regards,

Georg M. Schmölzer

Academic Editor

PLOS ONE

Journal Requirements:

Reviewers' comments:

Reviewer's Responses to Questions

**Comments to the Author**

1. Is the manuscript technically sound, and do the data support the conclusions?

Reviewer #1: Yes

2. Has the statistical analysis been performed appropriately and rigorously? 

Reviewer #1: Yes

3. Have the authors made all data underlying the findings in their manuscript fully available?

Reviewer #1: Yes

4. Is the manuscript presented in an intelligible fashion and written in standard English?

Reviewer #1: Yes

5. Review Comments to the Author

Reviewer #1: Dear Authors,

Thank you very much for going through the trouble of doing research in the field of pediatric airway management. Manikin trials are mandatroy to evalaute new devices and new techniques for airway management but have their own limitations. I appreaciated that the authors mentioned some of these limitations in their discussion section of the manuscript. However, I was wondering what these data actually mean for the anesthesiologist? What are next meaningful steps to further evaluate the MultiViewScope? I suggest to include some information in the manuscript (discussion section) what these data mean and what the next steps could be.

Specidfic Comments:

1.) The MultiViewScope Stylet is not a comonly used device in the word. I suggest to include a picture in the manuscript and elaborate a little more the build of the design. This would help the readers that are not familiar with the device.

2.) Abstract, concusions: Please include information what the presented data actually mean. What is the concusion after performing this trial and presenting the data? The current version of the conclsion is a very short result summary.

3.) Introduction: ... pediatric airway is challenging... What makes the pediatric airway challenging? Please be more spcific in this section. Is it an anatmically challenging airway (Is it the visulization of the glottis? Tube positioning?)? Or a physilogically difficult airway (risk of hypoxia)? Or both?

4.) What age does the infant manikin mimiking Pierre Robin syndrome represent? Could it be a problem that potentiallt two different ages are used for this trial? Please comment.

5.) I am not sure if I understood the randomization process correctly. Why was the "normal" airway intubated first, followed by the Pierre Robing manikin? Where is the randomization with this approach?

6.) Data is reported as median? Statistics is reported as means? This seems confisuing, could you please report median with corresponding tests or means with the approbiate tests?

7.) Looking at Cormack and Lehane gradring and intubation times it seems that the difficult airway might not have been difficult. Could this have influenced the data presented? Please comment.

8.) Cormack Lehane grading looks VERY similar CL1 (1 to 1) vs CL2 (1 to 2) between devices in difficult airway scenario. Not sure how the p <0.001 fits to this? Is a CL1 vs CL2 clinically relevant?

9.) Discussion line 216 - 218. How can the MVS provide a better view then the Bonfils? Is there a published trial to support this? Most trials with the Bonfils describe a CL1 view (which is needed to advance the tube). Please check and comment.

6. PLOS authors have the option to publish the peer review history of their article (what does this mean?). If published, this will include your full peer review and any attached files.

Reviewer #1: **Yes: **Dr. Ruediger Noppens

---

## [Author Response · Author response to Decision Letter 0]

19 Jul 2020

Georg M. Schmölzer, M.D., Ph.D.

Academic Editor

PLOS ONE

19 July 2020

Dear Dr. Schmölzer,

Response to review of PONE-D-20-17747: Comparison of the MultiViewScope Stylet Scope and the direct laryngoscope with the Miller blade for the intubation in normal and difficult pediatric airways: a randomized, crossover, manikin study.

We would like to thank the reviewer for the comments and for the help in improving the quality and clarity of our manuscript. Below are our point-by-point responses to their questions and suggestions. We hope that our manuscript is now suitable for publication in PLOS ONE.

Reviewer's report:

Reviewer #1

General: 

Thank you very much for going through the trouble of doing research in the field of pediatric airway management. Manikin trials are mandatroy to evalaute new devices and new techniques for airway management but have their own limitations. I appreaciated that the authors mentioned some of these limitations in their discussion section of the manuscript. However, I was wondering what these data actually mean for the anesthesiologist? What are next meaningful steps to further evaluate the MultiViewScope? I suggest to include some information in the manuscript (discussion section) what these data mean and what the next steps could be.

Answer, as reviewer #1 suggested we include the information in the abstract and discussion.

Page 3, line 40: The results of this study mean that the MVS Stylet Scope may be a less invasive airway devise than the direct laryngoscope with the Miller blade in the pediatric airway management. For the next step, we need to evaluate the MVS Stylet Scope in the real patients as an observational study.

Page 25, line265: The results of this study mean that the MVS Stylet Scope may be a less invasive airway devise than the direct laryngoscope with the Miller blade in the pediatric airway management. For the next step, we need to evaluate the MVS Stylet Scope in the real patients as an observational study.

Major Compulsory Revisions (that the author must respond to before a decision on publication can be reached): 

1.) The MultiViewScope Stylet is not a comonly used device in the word. I suggest to include a picture in the manuscript and elaborate a little more the build of the design. This would help the readers that are not familiar with the device.

Response: as the reviewer #1 suggested we include the picture.

Page 7, line 92: Fig 1. The MVS Stylet Scope.

2.) Abstract, concusions: Please include information what the presented data actually mean. What is the concusion after performing this trial and presenting the data? The current version of the conclsion is a very short result summary.

Answer, as reviewer #1 suggested we include the information in the abstract and discussion.

Page 3, line 40: The results of this study mean that the MVS Stylet Scope may be a less invasive airway devise than the direct laryngoscope with the Miller blade in the pediatric airway management. For the next step, we need to evaluate the MVS Stylet Scope in the real patients as an observational study.

Page 25, line265: The results of this study mean that the MVS Stylet Scope may be a less invasive airway devise than the direct laryngoscope with the Miller blade in the pediatric airway management. For the next step, we need to evaluate the MVS Stylet Scope in the real patients as an observational study.

3.) Introduction: ... pediatric airway is challenging... What makes the pediatric airway challenging? Please be more spcific in this section. Is it an anatmically challenging airway (Is it the visulization of the glottis? Tube positioning?)? Or a physilogically difficult airway (risk of hypoxia)? Or both?

Answer, we added the sentence in the introduction.

Page 4, line 47: because of the following physiological features

4.) What age does the infant manikin mimiking Pierre Robin syndrome represent? Could it be a problem that potentiallt two different ages are used for this trial? Please comment.

Answer, the age of the Pierre Robin manikin is a 3–6-month.

Page 7, line 86: A 3–6-month

5.) I am not sure if I understood the randomization process correctly. Why was the "normal" airway intubated first, followed by the Pierre Robing manikin? Where is the randomization with this approach?

Answer, as the reviewer #1 suggested, we added the reason.

Page 7, line 112: We did not randomize the order of manikins, because we compared the devices not the manikins.

Page 25, line 257: Another limitation of this study was that we did not randomize the order of manikins. This may account for our result that the time of successful intubation in the difficult pediatric airway was short compared to the normal pediatric airway in the anesthesiolosy residents.

6.) Data is reported as median? Statistics is reported as means? This seems confisuing, could you please report median with corresponding tests or means with the approbiate tests?

Answer, as the reviewer #1 suggested, we added the explanations.

Page 9, line 137: Normally distributed data are shown in means and standard deviations (SD). Data, which were not normally distributed, are shown in medians and interpercentile ranges (IQR).

7.) Looking at Cormack and Lehane gradring and intubation times it seems that the difficult airway might not have been difficult. Could this have influenced the data presented? Please comment.

Answer, we believe that the small difference between the two manikins is due to the learning effects.

Page 24, line 239: It is noteworthy that the times of successful intubation and Cormack–Lehane scale are similar in the two manikins. The mean time of successful intubation in the difficult pediatric airway was 28.4 sec, which was shorter than that of the normal pediatric airway (30.7 sec) in the anesthesiology residents. This may be due to the learning effects, because the mean time of successful intubation in the difficult pediatric airway (28.0 sec) was longer than that of the normal pediatric airway (24.3 sec) in the experienced expert anesthesiologists. Hippard et al. have reported that the time to intubation in the difficult pediatric airway manikin was shorter than the normal pediatric airway manikins [20]. We do not consider the learning effects have influenced our results, because we compared the devises, not the manikins.

8.) Cormack Lehane grading looks VERY similar CL1 (1 to 1) vs CL2 (1 to 2) between devices in difficult airway scenario. Not sure how the p <0.001 fits to this? Is a CL1 vs CL2 clinically relevant?

Answer, since this study was a crossover study, the paired analysis was done. The paired analysis can detect small differences.

Page 22, line 212: Although the differences of Cormack–Lehane scale between the MVS Stylet Scope and the direct laryngoscope using Miller #1 blade were small, we consider the differences are clinically significant. The reason is that the participants in the both groups reported improved difficulty of intubation with the MVS Stylet Scope.

9.) Discussion line 216 - 218. How can the MVS provide a better view then the Bonfils? Is there a published trial to support this? Most trials with the Bonfils describe a CL1 view (which is needed to advance the tube). Please check and comment.

Response: as the reviewer #1 suggested, we deleted the sentence.

Page 23, line 231: Because of the difference in the tip, The MVS Stylet Scope might provide better view of vocal cords than Bonfils retromolar intubation fiberscope.

Minor Essential Revisions (such as missing labels on figures, or the wrong use of a term, which the author can be trusted to correct): 

None

Discretionary Revisions (which the author can choose to ignore): 

None

Sincerely yours,

Kohei Godai, M.D., Ph.D.

Department of Anesthesiology and Critical Care Medicine, Graduate School of Medical and Dental Sciences, Kagoshima University, 8-35-1 Sakuragaoka, Kagoshima 890-8520, Japan

Tel: +81-99-275-5430; Fax: +81-99-265-1642

E-mail: kxg179@icloud.com

---

## [Editor Report · Decision Letter 1]

30 Jul 2020

Comparison of the MultiViewScope Stylet Scope and the direct laryngoscope with the Miller blade for the intubation in normal and difficult pediatric airways: a randomized, crossover, manikin study

PONE-D-20-17747R1

Dear Dr. Godai,

We’re pleased to inform you that your manuscript has been judged scientifically suitable for publication and will be formally accepted for publication once it meets all outstanding technical requirements.

Kind regards,

Georg M. Schmölzer

Academic Editor

PLOS ONE
---

## [Editor Report · Acceptance letter]

5 Aug 2020

PONE-D-20-17747R1 

Comparison of the MultiViewScope Stylet Scope and the direct laryngoscope with the Miller blade for the intubation in normal and difficult pediatric airways: a randomized, crossover, manikin study 

Dear Dr. Godai:

I'm pleased to inform you that your manuscript has been deemed suitable for publication in PLOS ONE. Congratulations! Your manuscript is now with our production department. 

Kind regards, 

on behalf of

Dr. Georg M. Schmölzer 

Academic Editor

PLOS ONE